# Ecological Momentary Assessment of Awake Bruxism Behaviors: A Scoping Review of Findings from Smartphone-Based Studies in Healthy Young Adults

**DOI:** 10.3390/jcm12051904

**Published:** 2023-02-28

**Authors:** Anna Colonna, Alessandro Bracci, Jari Ahlberg, Mariana Barbosa Câmara-Souza, Rosaria Bucci, Paulo César Rodrigues Conti, Ricardo Dias, Alona Emodi-Perlmam, Riccardo Favero, Birgitta Häggmän-Henrikson, Ambrosina Michelotti, Laura Nykänen, Nikola Stanisic, Efraim Winocur, Frank Lobbezoo, Daniele Manfredini

**Affiliations:** 1Department of Biomedical Technologies, School of Dentistry, University of Siena, 53100 Siena, Italy; 2Department of Neurosciences, School of Dentistry, University of Padova, 35128 Padova, Italy; 3Department of Oral and Maxillofacial Diseases, University of Helsinki, 00100 Helsinki, Finland; 4Ingá University Center, Maringá 87035-510, Brazil; 5Department of Neurosciences, Reproductive Sciences and Oral Sciences, Section of Orthodontics and Temporomandibular Disorders, University of Naples “Federico II”, 80138 Naples, Italy; 6Bauru School of Dentistry, University of Sao Paulo, Sao Paulo 05508-000, Brazil; 7Institute of Oral Implantology and Prosthodontics, Dentistry Department, Faculty of Medicine, University of Coimbra, 3004-531 Coimbra, Portugal; 8Department of Oral Rehabilitation, The Maurice and Gabriela Goldschleger School of Dental Medicine, Sackler Faculty of Medicine, Tel Aviv University, Tel Aviv 69978, Israel; 9Department of Orofacial Pain and Jaw function, Faculty of Odontology, Malmö University, 211 19 Malmö, Sweden; 10Department of Orofacial Pain and Dysfunction, Academic Centre for Dentistry Amsterdam (ACTA), University of Amsterdam and Vrije Universiteit Amsterdam, 1081 HV Amsterdam, The Netherlands

**Keywords:** bruxism, awake bruxism, ecological momentary assessment, masticatory muscle activity

## Abstract

Background: The recent introduction of ecological momentary assessment (EMA) smartphone-based strategies has allowed achieving some interesting data on the frequency of different awake bruxism (AB) behaviors reported by an individual in the natural environment. Objective: The present paper aims to review the literature on the reported frequency of AB based on data gathered via smartphone EMA technology. Methods: On September 2022, a systematic search in the Pubmed, Scopus and Google Scholar databases was performed to identify all peer-reviewed English-language studies assessing awake bruxism behaviors using a smartphone-based Ecological Momentary Assessment. The selected articles were assessed independently by two authors according to a structured reading of the articles’ format (PICO). Results: A literature search, for which the search terms “Awake Bruxism” and “Ecological Momentary Assessment” were used, identified 15 articles. Of them, eight fulfilled the inclusion criteria. The results of seven studies using the same smartphone-based app reported a frequency of AB behaviors in the range between 28.3 and 40% over one week, while another investigation adopted a different smartphone-based EMA approach via WhatsApp using a web-based survey program and reported an AB frequency of 58.6%. Most included studies were based on convenience samples with limited age range, highlighting the need for more studies on other population samples. Conclusions: Despite the methodological limits, the results of the reviewed studies provide a standpoint for comparison for future studies on the epidemiology of awake bruxism behaviors.

## 1. Introduction

Bruxism is an oral condition that is gaining increasing attention in several disciplines, such as dentistry, psychology, neurology, and sleep medicine.

Recently, a panel of experts provided separate definitions for awake bruxism (AB) and sleep bruxism (SB).

Awake Bruxism (AB) is currently defined as a masticatory muscle activity (MMA) during wakefulness that is characterized by repetitive or sustained tooth contact and/or by bracing or thrusting of the mandible and is not a movement disorder in otherwise healthy individuals [1].

Sleep bruxism (SB) is a masticatory muscle activity during sleep that is characterized as rhythmic (phasic) or non-rhythmic (tonic) and is not a movement disorder or a sleep disorder in otherwise healthy individuals [1].

In accordance with these definitions, AB must be distinguished from SB, which may have different etiology, comorbidities, and consequences because of the different spectrum of muscle activities exerted in relation to the different circadian manifestation.

The prevalence of AB has been reported to be up to 30% across populations, while SB prevalence has been reported to be 6–8% [2]. Nevertheless, until now, most research has focused on SB, while knowledge of AB is fragmental, and less literature data on AB are available compared to SB [2]. There are few epidemiological data on AB, and the findings are not easy to summarize due to the adoption of different assessment strategies. In addition, most of the information on AB prevalence has been reported from research on single-observation point studies [1,2,3,4,5].

Consequently, as part of the works that led to the definition of the Standardized Tool for the Assessment of Bruxism (STAB), [6] the use of ecological momentary assessment (EMA) strategies to report AB behaviors has been recommended and adopted in several studies [1,2,3,6,7,8,9,10,11,12,13,14,15,16,17,18,19,20,21,22], based on its potential usefulness as a simple method to collect real-time data in the natural environment. Such procedure is also referred to as experience sampling method (ESM) and requires a real-time report of behaviors or feelings, in relation to whichever condition is being studied (e.g., AB behaviors) [21]. In short, over a time frame within the course of daily affairs, an individual is prompted at fixed or random time points to answer questions about what he/she is currently doing and/or experiencing. Doing so, multiple recording points during the day, close in time to the experience in the natural environment, are allowed [22].

Over the years, EMA approaches have been introduced in the field of AB investigations [23,24], with a recent focus on the development of smartphone-based strategies to collect data in the clinical and research settings. By using this strategy, a patient receives information on the smartphone application to use and the instructions on how to use it. Upon receiving an alert, the subjects have to focus on their current oral condition and tap on the corresponding display icon (e.g., relaxed jaw muscles, teeth contact, teeth clenching, teeth grinding, mandible bracing) [7,8]. Such approach allows the collection of data regarding the frequency of the different AB activities reported by an individual in the natural environment. Nonetheless, despite the potential advantages of using this strategy being quite intuitive, also as far as patients’ education purposes are concerned, there are so far only little available data. Studies over a one-week period described a quite wide range of average frequency values for AB behaviors in otherwise healthy young adults [8,10,11,12,13,14,19]. Concerning the specific behaviors, teeth contact is the most common finding, whilst the report of teeth grinding is almost absent. This approach is in line with the need to collect as much data as possible to understand which degree of AB behaviors may become harmful, if any [25,26].

In view of this, it is important to summarize the current findings in this emerging research field to provide a basis for comparisons as well as possible suggestions for future research. Within these premises, a scoping review was performed to report on AB prevalence data gathered via smartphone-based EMA technology.

## 2. Materials and Methods

### Search Strategy and Selection Criteria

On 10 September 2022, a systematic search of the literature was performed to identify all peer-reviewed English-language citations that were relevant to the review topic, viz., smartphone-based Ecological Momentary Assessment of Awake Bruxism.

As a first step, a search using Medical Subjects Headings (MeSH) terms in the National Library of Medicine Medline (PubMed) database was performed by using the query “Awake Bruxism” AND “Ecological Momentary Assessment” to identify a list of potential papers for inclusion. As a next step, the same strategy was adopted to identify papers in the Scopus and Google Scholar databases. In addition, search expansion strategies were adopted to identify any additional potentially relevant citation (i.e., related articles, hands-on search in private libraries, reference lists of the included articles). The literature search was limited to all the articles on adult populations (>18 years). The inclusion criteria were limited to: (1) studies written in English, (2) papers focused on the frequency of AB evaluated via a smartphone-based EMA approach.

The selected articles were read according to a PICO-like structured strategy (i.e., Population/Intervention/Comparison/Outcome). The population (“P”) was described in terms of sample size, inclusion criteria, and demographic characteristics. The intervention (“I”) concerned information on the study design, the assessment approach, the number and qualification of the examiners and the statistical analysis. The comparison (“C”) included data on the control group depending on the study design, if it is present. The outcome (“O”) was reported in terms of AB frequency data. The main conclusions of each study’s authors were also included. 

Two of the authors (A.C., A.B.) independently reviewed the titles and abstracts of all articles and performed a full-text eligibility check. The articles that were potentially meeting the inclusion criteria for the review were retrieved in full text. In all cases of doubt regarding the potential inclusion of an article or data interpretation, the main supervisor (D.M.) was involved. All the leading authors of the included investigations were contacted to join the review team and contribute to the expansion of the research strategy in the additional steps. Each of them also contributed with a handmade search in his/her own university library catalogue. Once the review team agreed on the articles included in the review, the two main reviewers performed data extraction based on the above-described PICO strategy. Some authors involved in the STAB preparation were also invited to contribute to the literature review discussion.

## 3. Results

### 3.1. Overview

The literature search identified 15 articles. Title and abstract reading led to the exclusion of two articles, which were clearly not relevant [20,27]. The full text for the remaining 13 articles was retrieved. Of these, five were excluded [7,9,15,16,17] for not fulfilling the inclusion criteria, (i.e., the reason for exclusion was No information about AB frequency); thus, a total of eight papers [8,10,12,13,14,16,18,19] were included in the review (Figure 1).

The included papers covered a wide spectrum of populations of different sex, age and ethnic background. The studies were performed on subjects living in Italy, Portugal, Israel and Brazil. The age of the subjects varied from 19 to 35 years, and the sample size ranged from 30 to 151. Regarding the sex distribution, a predominance of females was found.

### 3.2. Summary of the Studies

Several studies had common methodological features. In most cases, the study sample was represented by students and/or by a restricted age group, and the majority of the participants were females (Table 1).

In all studies [8,10,12,13,14,16,18,19], in order to evaluate AB behaviors, the participants had to answer (i.e., EMA) by tapping on the display within 5 min from the alert on the current condition of their jaw muscles, and they were monitored over a one-week period; in some cases, intervals of several weeks were also evaluated [10,12]. All studies [8,10,12,13,14,16,19] except one [18] adopted the same smartphone-based application.

The findings in the eight papers suggested that the frequency of AB behaviors report (i.e., the percentage of positive alerts over one week) in young adults was within the 28.3–58.6% range considering the sum of tooth contact, teeth clenching, teeth grinding, mandible bracing frequency. In detail, the seven studies that used the same smartphone-based app (BruxApp^®^, WMA Srl., Florence, Italy) reported an AB frequency range between 28.3 and 40% [8,10,12,13,14,16,19], while another investigation adopted a different smartphone-based EMA approach via WhatsApp using a web-based survey program called Mentimeter^®^ and reported an AB frequency of 58.6% (Table 2, Figure 2A–E and Figure 3A–E) [18].

Among the different AB behaviors, teeth contact appeared to be the most common behavior. Report of teeth grinding was almost absent, as shown in Table 2, Figure 2A–E and Figure 3A–E.

## 4. Discussion

A major concern for current bruxism knowledge is the paucity of literature data on the epidemiology of awake bruxism with respect to sleep bruxism [2]. A common suggestion from several reviews is that an improvement of such knowledge would be helpful to clarify its clinical relevance [26,28,29,30,31].

Considering that bruxism is a masticatory muscle activity, the recommended strategy is to have electromyographic recordings of the jaw muscles during wakefulness. Nonetheless, performing an hour-long EMG recording of jaw muscle activity during wakefulness is difficult due to potentially poor patient compliance, as well as for technical reasons [1,2]. Consequently, to date, the information on AB prevalence is mainly based on retrospective self-reported data collections at a single observation point, with questionnaire approaches adopted in both clinical and research settings. Such a strategy requires an individual to recollect the frequency of a habit over the timespan covered by the report, thus representing a potential risk of bias. In addition, the intensity and duration of specific masticatory muscle activities cannot be quantified via a self-report [1]. It is also interesting to note that, currently, there are no universally adopted questionnaires for the assessment of AB [20]. The most frequent approach provides the use of AB items included in history using instruments that were designed for broader scopes, such as the report of temporomandibular disorders (e.g., Diagnostic Criteria for Temporomandibular Disorders (DC/TMD) [32], oral behaviors (e.g., Oral Behaviors Checklist) [33] and bruxism in general (e.g., Bruxscale) [34].

As recently suggested by several studies [8,9,10,11,12,13,14,15,16,17,18,19,20] and by the expert consensus papers on bruxism definition [1,3], these limitations may be overcome by adopting EMA strategies, which are based on the report of a condition in real time. Such an approach is not a novelty in the field of psychological sciences [23] and has gained popularity also in medical science [35,36,37]. The application of the EMA principles to the study of AB has emerged as an interesting assessment option, since it is a simple method to collect real-time data in the natural environment. The EMA approach may be optimized with the use of smartphone apps, which are easy to use and intuitive [38,39]. The smartphone-based EMA approach is carried out in the natural setting and may be prolonged for several days, thus offering a potential advantage in terms of ecological validity, even compared with hour-long EMG recordings during wakefulness. In detail, it allows for the real-time collection of self-report data on the momentary presence of different oral conditions (e.g., relaxed jaw muscles, tooth contact, teeth clenching, teeth grinding, mandible bracing) that are related to the spectrum of AB activities. This approach allows monitoring AB behaviors over time and testing for potential Ecological Momentary (EMI) Intervention effects.

In this view, it is important to summarize the current knowledge on AB to provide a basis for comparison as well as possible suggestions for future studies. Based on that, a literature review was executed to report on AB prevalence data gathered via smartphone-based EMA technology.

Only eight papers met the inclusion criteria [8,10,12,13,14,16,18,19]. A major concern emerging from the literature overview is that many studies dealing with the EMA assessment of AB demonstrated a potentially poor external validity of their findings. Many studies were performed in convenience samples of non-representative populations, which is the main reason for our decision to avoid any qualitative and quantitative assessment of the findings. In fact, most investigations were performed on a study sample recruited exclusively from university students [8,10,12,13,14,16], with the exception of Pereira et al. [18] and Bucci et al. [19] studies. It is also important to point out that the studies that met the inclusion criteria originated from the same group of researchers, with some variations. Such flaws may affect the external validity of the findings and the consistency of the prevalence data across the studies. From a methodological viewpoint, taking into account the above considerations, the findings of the present review should be interpreted with caution and be viewed as a narrative standpoint for future comparisons.

In general, the papers included in the final review [8,10,12,13,14,16,18,19] reported an average AB frequency behavior over a one-week period within the range of 23.6–58.6% for otherwise healthy young adults. Nonetheless, it is important to underline that all studies, except Pereira et al.’s [18], used the same study protocol, and the prevalence of AB among these articles was much more consistent, varying from 23.6% to 40.3%. In the study by Pereira et al. [18], which instead used a different method based on WhatsApp messages, the prevalence of AB behaviors was higher (58.6%). In fact, all but one study [8,10,12,13,14,16,19] used the same dedicated smartphone application, which sends alerts at randomly during the day to collect data on self-reported AB. In this case, the subjects must answer (i.e., EMA) by tapping on the display icon that corresponds to the current condition of his/her jaw muscles: relaxed jaw muscles; teeth contact; teeth clenching; teeth grinding; mandible bracing (i.e., jaw clenching without teeth contact). On the other hand, Pereira et al. [18] used a web-based survey program called Mentimeter^®^. For this program the possible responses are as follows: (a) I am not touching my teeth; (b) I am not touching my teeth, but I feel my muscles are contracted; (c) I am slightly touching my teeth; (d) I am clenching my teeth; or (e) I am grinding my teeth. It is interesting to note how the relaxed jaw muscle condition is absent amongst the potential options. This between-study difference could explain the higher percentage of AB behaviors compared to the other studies, in addition to other potential explanations concerning differences in the socio-cultural aspects of the study populations.

In all studies, the subject answered to an alert, i.e., EMA, by tapping on the smartphone display within 5 min. The monitoring period was 7 consecutive days for all studies, but the investigation by Pereira et al. [18] was based on 10 alerts/day instead of the 20 alerts at random intervals used in the other studies. It is important to point out that the exclusion of subjects with signs or symptoms of TMD or other painful chronic disorders based on the recommendations of the Diagnostic Criteria for TMD (DC/TMD) [32] is a common feature of all studies.

It is interesting to note that, among the different AB behaviors, teeth contact was the most frequently reported behavior in all studies, whilst teeth clenching was much less frequently reported than commonly believed, and the report of teeth grinding was almost absent. Concerning sex differences, four studies examined this aspect and did not find any consistent pattern for the differences between AB prevalence in males and females. On the other hand, Zani et al. [14] found that the mean of the relaxed jaw muscles condition was higher in males than in females. Câmara-Souza et al. [13] underlined that significantly higher AB values were found in females, who had approximately 1.5 times more AB episodes than males. In general, these findings point towards a potentially higher report of AB in females.

Given that the data collection span covered one week, a potential EMI (Ecological Momentary Intervention) effect may also be considered. This is demonstrated by the fact that the use of the application over time leads to a decrease in AB behaviors, as highlighted by Zani et al. and Dias et al. [12,14]. The authors adopted a study design with multiple monitoring weeks and noticed that the percentage of AB reports went down from 38% to 26% and from 37.5% to 31%, respectively. This may also suggest potentially therapeutic opportunities. EMI based on the biofeedback mechanism has already been used effectively in people who show potentially dangerous behaviors as a strategy to recognize and modify them [10,40]. The use of technology has introduced a new possible way for clinicians to engage patients from a therapeutic viewpoint, which may be adopted for AB as well, when required. In this scenario, myofascial pain patients with anxiety personality and stress sensitivity are theoretically the ideal target for the app-based EMI approach, due to the potential influence on emotion-related mandible bracing [10].

Future epidemiological studies should carefully avoid the selection of non-representative populations and recruit large study samples. On the other hand, it is fundamental that future investigations are based on carefully organized and standardized training sessions addressing the issue of patients’ comprehension, as suggested by Nykänen et al. [17]. Indeed, poor patients’ compliance and comprehension, along with the presenting clinician’s untrained skills, may represent methodological flaws and sources of bias hampering the generalization of the findings.

Concerning the above, the current body of evidence is of low quality because it is mainly based on a single smartphone application to record a real-time report on AB behaviors, and therefore generalization will only be possible in the future with the use of different EMA methods and the involvement of as many other investigators as possible.

The findings presented in this review could be seen as a reference point for future investigations on the epidemiological features of AB. Future EMA research on the frequency of AB activities in healthy individuals and on their additive frequency in selected populations with comorbid conditions (e.g., psychological and social impairment, orofacial pain, sleep disorders) may contribute to a better understanding of this complex topic. The EMA approach—which is also called experience sampling methodology (ESM)—might be used in association with other strategies, as suggested in the STAB [6]. A combination of the different approaches may represent the best possible strategy to overcome the limitations of the different stand-alone approaches.

## 5. Conclusions

The present review assessed the literature on the report of data on awake bruxism behaviors gathered via smartphone-based EMA technology. The reviewed articles have some methodological limits, such as convenience samples and a limited age range. As part of this literature review, some suggestions to improve the standardization of data collection are presented, based on the current bruxism construct and the proposals by the international expert panel that worked on the Standardized Tool for the Assessment of Bruxism.

The results of seven studies using the same smartphone-based app reported a frequency of AB behaviors in the range between 28.3 and 40% over one week, while another investigation adopted a different smartphone-based EMA approach via WhatsApp using a web-based survey program and reported an AB frequency of 58.6%.

## Figures and Tables

**Figure 1 jcm-12-01904-f001:**
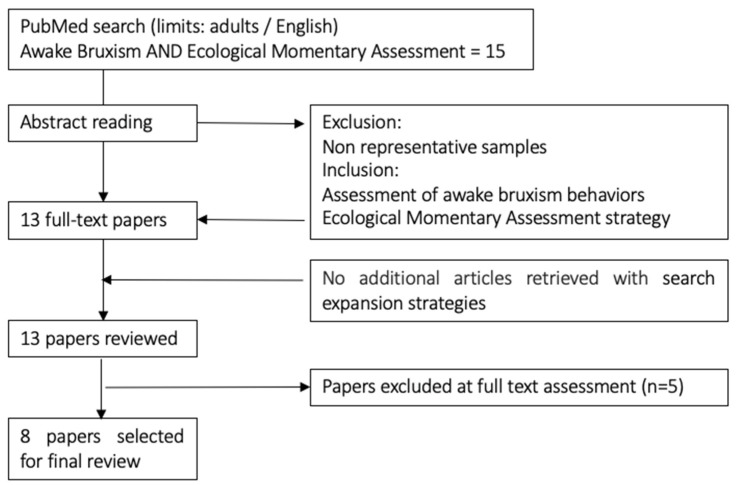
Literature search strategy. Different steps and criteria for the selection of papers.

**Figure 2 jcm-12-01904-f002:**
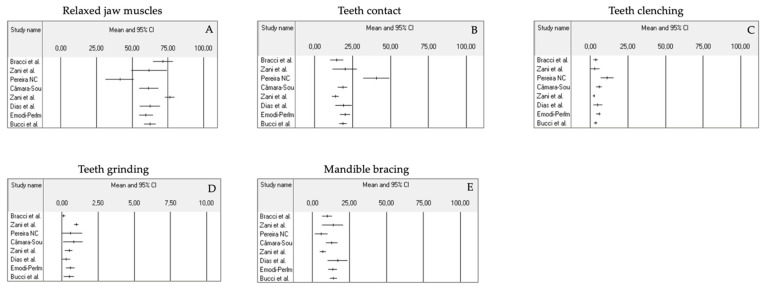
(**A**–**E**) Mean values distribution of the frequency data and confidence interval for AB behaviors (e.g., relaxed jaw muscles, teeth contact, teeth clenching, teeth grinding, mandible bracing)—comparison among studies [8,10,12,13,14,16,18,19].

**Figure 3 jcm-12-01904-f003:**
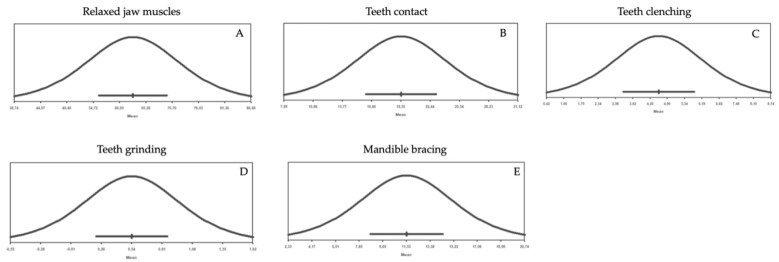
(**A**–**E**) Reported frequency data of AB behaviors (e.g., relaxed jaw muscles, teeth contact, teeth clenching, teeth grinding, mandible bracing).

**Table 1 jcm-12-01904-t001:** Summary of the findings from studies included in the final review. PICO Table.

Study’s First Author and Year	Population (Patients/Problem)	Intervention	Comparison (Control Group)	Outcome
Bracci et al., 2018 [8]	N = 46 (26 F, 20 M; m.a. 24.2 ± 1.7) Italy University students	Ecological momentary assessment using a dedicated smartphone application to record a real-time report on AB behaviors	Not applicable	AB frequency: 28.3%
Zani et al., 2019 [10]	N = 30 (21 F, 9 M; 24 ± 3.5) Italy University students	Ecological momentary assessment using a dedicated smartphone application to record a real-time report on AB behaviors	Not applicable	AB frequency: 38%
Pereira NC et al., 2020 [18]	N = 38 (15 F, 23 M; m.a. 22.1) Brazil Subjects representative of the general population	Ecological momentary assessment using an online device (mentimeter) to record a real-time report on AB behaviors	Not applicable	AB frequency: 58.6%
Câmara-Souza et al., 2020 [13]	N = 69 (50 F, 19 M; m.a. 18.6 ± 1.5) Brazil University students	Ecological momentary assessment using a dedicated smartphone application to record a real-time report on AB behaviors	Not applicable	AB frequency: 38.4%
Zani et al., 2021 [14]	N = 153 (93 F, 60 M; m.a. 22.9 ± 3.2) Italy University students	Ecological momentary assessment using a dedicated smartphone application to record a real-time report on AB behaviors	Not applicable	AB frequency: 23.6
Dias et al., 2021 [12]	N = 31 (27 F, 4 M; a.r. 20–24) Portugal University students	Ecological momentary assessment using a dedicated smartphone application to record a real-time report on AB behaviors	Not applicable	AB frequency: 37.5%
Emodi-Perlman et al., 2021 [16]	N = 106 (67 F, 39 M; m.a. 24.4 ± 2.99) Israel University students	Ecological momentary assessment using a dedicated smartphone application to record a real-time report on AB behaviors	Not applicable	AB frequency: 40.3%
Bucci et al., 2022 [19]	N = 151 (99 F, 52 M; m.a. 27.2 ± 8.1) Italy Subjects representative of the general population	Ecological momentary assessment using a dedicated smartphone application to record a real-time report on AB behaviors	Not applicable	AB frequency: 37.5%

m.a., mean age; a.r., age range; AB, awake bruxism.

**Table 2 jcm-12-01904-t002:** Mean values of the frequency data (percentage of positive answers for the different AB behaviors over the 7-day observation period)—comparison among studies.

	Bracci et al., 2018 [8]	Zani et al., 2019 T1 [10]	Zani et al., 2019 T2 [10]	Pereira NC et al., 2020 [18]	Câmara-Souza et al., 2020 [13]	Zani et al., 2021 [14]	Dias et al., 2021 T1 [12]	Dias et al., 2021 T2 [12]	Dias et al., 2021 T3 [12]	Emodi-Perlman et al., 2021 [16]	Bucci et al., 2022 [19]
	N = 46	N = 30	N = 30	N = 38	N = 69	N = 153	N = 31	N = 31	N = 31	N = 106	N = 151
Activity	Mean frequency	Mean frequency	Mean frequency	Mean frequency	Mean frequency	Mean frequency	Mean frequency	Mean frequency	Mean frequency	Mean frequency	Mean frequency
Relaxed jaw muscles	71.7	62	74	41.4	61,6	76.4	62.5	67.8	69.0	59.7	62.5
Teeth contact	14.5	20	11	40.9	18.6	13.6	19.1	16.2	14.7	20.2	18.8
Mandible bracing	10.0	14	13	5.9	13.1	7.0	17.2	16.9	15.8	13.7	14.3
Teeth clenching	3.7	3	2	11.2	5.9	2.5	4.9	2.3	1.9	5.7	3.6
Teeth grinding	0.1	1	1	0.6	0.8	0.5	0.3	0.3	0.3	0.6	0.5
Gender differences	No	No	No	No	Yes	Yes	///	///	///	No	///

## Data Availability

Not applicable.

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
