# Peer review of "Ecological Momentary Assessment of Awake Bruxism Behaviors: A Scoping Review of Findings from Smartphone-Based Studies in Healthy Young Adults"

_jcm, 2023, doi:10.3390/jcm12051904_

Round 1

Reviewer 1 Report

Dear Authors,

Although it is an interesting topic, it lacks scientific rigour.

Looks like a mini review - only 40 references. It should be about or more than 100 references for a review paper.

Tab1.  Demographic data of the study population have some gaps in age range.

The manuscript is well written. However, I am not sure which category it should be in. It stated as mini review

Best Regards

Author Response

Dear Reviewer,

thank you for the interesting comments and suggestions.

Please find below a point-by-point answer.

Dear Authors,

Although it is an interesting topic, it lacks scientific rigour.

Looks like a mini review - only 40 references. It should be about or more than 100 references for a review paper.

The references are only 40 as currently the studies published on this topic are not yet numerous as it has only been introduced in the clinical and research fields for a few years. This means that additional references can be added just to increase the number of papers cited in the discussion.

Tab1.  Demographic data of the study population have some gaps in age range.

Thanks for pointing this issue out. Gaps in age range are present because most of the articles report the mean age but not the age range.

The manuscript is well written. However, I am not sure which category it should be in. It stated as mini review.

Thanks for appreciating our writing style. We leave the decision about the best way to label the manuscript up to the editorial board.

We really hope our comments are appreciated and look forward to your feedback.

Sincerely Yours

The authors

Reviewer 2 Report

This study is a literature review that assessed prevalence of AB from smartphone-based studies 3

in healthy young adults. Abstract section should be rewrite. Most of abstract section was background, no methods, no results and no conclusion.  

 Introduction:

Its mandatory to briefly define ecological momentary assessment in introduction for those reader who will read these terms for the first time.

The authors did not distinguish if this review is scoping or systematic review. its mandatory to decide and clearly state that its scoping review. In the title, scoping review should be added.

Authors should clearly state aims of study

Authors should state which checklist or guidelines they used in this review such as PRISMA?  then authors should follow this guideline

The authors did not clearly state the inclusion criteria based on PICOS criteria. authors should definitely state type of patients/interventions/comparator/outcomes and study design

Subheadings of methods and results should be conforming the PRISMA checklist

The authors have never mentioned inclusion criteria and exclusion criteria

Authors should clearly state the outcomes of interest they assessed in this review, and by quantitatively an analysis presented the results    

Author should use a charting or visualization chart  to report and present the results (scoping review)

The authors did not assess the quality of study and the evidence provided by this study

Author Response

Dear Reviewer,

thank you for the interesting comments and suggestions.

Please find below a point-by-point answer.

This study is a literature review that assessed prevalence of AB from smartphone-based studies 3

in healthy young adults. Abstract section should be rewrite. Most of abstract section was background, no methods, no results and no conclusion.  

Based on your suggestions, the text has been reviewed.

 Introduction:Its mandatory to briefly define ecological momentary assessment in introduction for those reader who will read these terms for the first time.

The definition of ecological momentary assessment (EMA) is present in the lines 77-83 of the introduction. Based on your suggestion, we have made it clearer in the revised text.

The authors did not distinguish if this review is scoping or systematic review. its mandatory to decide and clearly state that its scoping review. In the title, scoping review should be added.

Thanks for the remark. We have changed the title accordingly.

Authors should clearly state aims of study

As specified in the text, the purpose of this scoping literature review, executed to report on AB prevalence data gathered via smartphone-based EMA technology, is to summarize current knowledge on AB and to summarize current findings in this emerging research field to provide a basis for comparison as well as possible suggestions for future research.

Authors should state which checklist or guidelines they used in this review such as PRISMA?  then authors should follow this guideline

The authors did not clearly state the inclusion criteria based on PICOS criteria. authors should definitely state type of patients/interventions/comparator/outcomes and study design

Thanks for the remark. We have changed the manuscript accordingly.

Subheadings of methods and results should be conforming the PRISMA checklist

The authors have never mentioned inclusion criteria and exclusion criteria

The inclusion and exclusion criteria are presented in the paragraph "Search Strategy and Selection Criteria".

Authors should clearly state the outcomes of interest they assessed in this review, and by quantitatively an analysis presented the results    

Author should use a charting or visualization chart to report and present the results (scoping review)

The authors did not assess the quality of study and the evidence provided by this study

Thanks for the comment. In this case a quantitative analysis is not possible because the meta-analysis of the data would presuppose homogeneity of the data collection.

In any case, based on your suggestions, the text has been reviewed.

We really hope our comments are appreciated and look forward to your feedback.

Sincerely Yours

The authors

Reviewer 3 Report

Comments:

A) Authors need to provide as a tabular column  of 8 papers selected for final review

Author Response

Dear Reviewer,

thank you for the interesting comments and suggestions.

Please find below a point-by-point answer.

Authors need to provide as a tabular column of 8 papers selected for final review

As you suggested, we have added the requested information.

We really hope our comments are appreciated  and look forward to your feedback.

Sincerely Yours

The authors

Round 2

Reviewer 2 Report

Unfortunately,  authors dud not addressed my comments appropriately.  Since my comments concerned a major methodological issues, I must reject the manuscript in its current state 

Author Response

Thanks for the reply.

Please find below a point-by-point answer.

"Unfortunately,  authors dud not addressed my comments appropriately.  Since my comments concerned a major methodological issues, I must reject the manuscript in its current state."

We edited the text again, and based on your suggestions we made the following new changes:
-We edited the manuscript in scoping review.
- We added "PICO" table.
- We explained why it is not possible to carry out qualitative-quantitative evaluation.
- We refined the text with a mother tongue.
- We detailed the results more appropriately.

We look forward to specific suggestions and remain at your disposal.
Thanks in advance